# Pleiotropic genetic architecture and novel loci for C-reactive protein levels

Fotios Koskeridis [1] ✉, Evangelos Evangelou[1,2,3], Saredo Said[4], Joseph J. Boyle[5], Paul Elliott [3,6,7], Abbas Dehghan [3,6,7] & Ioanna Tzoulaki [1,2,3,6,7]

C-reactive protein is involved in a plethora of pathophysiological conditions. Many genetic loci associated with C-reactive protein are annotated to lipid and glucose metabolism genes supporting common biological pathways between inflammation and metabolic traits. To identify novel pleiotropic loci, we perform multi-trait analysis of genome-wide association studies on C-reactive protein levels along with cardiometabolic traits, followed by a series of in silico analyses including colocalization, phenome-wide association studies and Mendelian randomization. We find 41 novel loci and 19 gene sets associated with C-reactive protein with various pleiotropic effects. Additionally, 41 variants colocalize between C-reactive protein and cardiometabolic risk factors and 12 of them display unexpected discordant effects between the shared traits which are translated into discordant associations with clinical outcomes in subsequent phenome-wide association studies. Our findings provide insights into shared mechanisms underlying inflammation and lipid metabolism, representing potential preventive and therapeutic targets.

Metabolic and immune are evolutionarily conserved and functionally entangled systems whose interplay is believed to play a central role in the pathophysiology of a wide spectrum of cardiovascular diseases[1]. As a result, metabolic disturbances are commonly associated with immune responses in relation to cardiovascular outcomes. For example, high low-density lipoprotein (LDL) cholesterol levels and their subsequent modification are involved in the chronic inflammatory response of the vascular wall[2]. C-reactive protein (CRP), a marker of systemic inflammation, is an acute phase reactant secreted mainly by the liver and released in high concentrations in blood as a protective response to harmful irritants such as pathogens or damaged tissue. Genetic and environmental determinants of CRP levels have been widely studied as its levels have been associated with a plethora of cardiometabolic phenotypes and its causal role has been under investigation for years[3].

Genome-wide association studies (GWAS) on CRP levels have identified numerous, robustly associated, genetic loci associated with CRP levels[4] supporting a polygenic model for this trait. Several of the identified CRP loci are annotated not only to inflammation-related genes but also to lipid and glucose metabolism-related genes whereas several have been identified as pleiotropic affecting both phenotypes[5,6] providing further evidence towards common biological pathways between inflammation and cardiometabolic traits. A deeper understanding of the interplay between inflammation and cardiovascular risk factors is likely to highlight important disease pathways and interventions with added clinical benefit.

Here, we investigate further the pleiotropic nature of the genetic architecture of CRP in relation to cardiometabolic traits. We performed a Multi-Trait Analysis of GWAS (MTAG) on CRP levels with established cardiometabolic risk factors followed by a series of in silico

[1]Department of Hygiene and Epidemiology, University of Ioannina Medical School, Ioannina, Greece. [2]Institute of Biosciences, University Research Center of Ioannina, University of Ioannina, 45110 Ioannina, Greece. [3]Department of Epidemiology and Biostatistics, School of Public Health, Imperial College London, London, UK. [4]Nuffield Department of Population Health, University of Oxford, Oxford, UK. [5]National Heart and Lung Institute, Imperial College London, London, UK. [6]UK Dementia Research Institute, Imperial College London, London, UK. [7]BHF Centre of Excellence, Imperial College London, London, UK. ✉e-mail: f.koskeridis@uoi.gr

analyses to identify novel pleiotropic genes, their tissue site of action, and evidence for causal associations with a range of disease outcomes. Ultimately, the study provides additional biological insights into low-grade inflammation and highlights biological pathways that are likely to link inflammation to different cardiometabolic diseases.

## Results

### Multi-trait GWAS and novel CRP genetic loci

A schematic overview of our study is shown in Fig. 1. We performed multivariate MTAG between CRP and five cardiometabolic risk factors including high-density lipoprotein (HDL) levels, low-density lipoprotein (LDL) levels, triglyceride (TG) levels, body mass index (BMI), and cigarettes per day (CPD), as well as bivariate MTAG between CRP and each cardiometabolic trait separately. Multivariate MTAG identified 797 independent signals in 283 genetic loci associated with CRP at genome-wide significance (GWS) level ($P < 5 \times 10^{-8}$) [Supplementary Data 1, 2, Fig. 2]. For the remaining traits, we found 549 independent signals (185 loci) for HDL, 527 (144) for LDL, 534 (173) for TG, 1,552 (740) for BMI, and 108 (62) for CPD [Supplementary Data 3, Fig. 2]. Of the 797 CRP SNPs, 295 (151 loci) were associated with at least another examined trait ($P < 5 \times 10^{-8}$), and 8 of them (8 loci) with at least 4 of the 5 examined traits [Supplementary Data 4, Supplementary Fig. 1].

The bivariate MTAG between CRP and each of the examined risk factors identified 41 additional loci for CRP (Supplementary Data 5). The 324 CRP loci (283 from multivariate plus 41 from bivariate MTAG) corresponded to 41 novel genomic loci for CRP (Table 1, Supplementary Figs. 2, 3). The strongest novel CRP locus was in the *LPL* gene

(rs35237252, $P = 5.1 \times 10^{-20}$) that encodes lipoprotein lipase which is expressed in heart, muscle, and adipose tissue[7].

### Functional annotation and pathway enrichment

We used ANNOtate VARiation (ANNOVAR)[8] and multi-marker analysis of genomic annotation (MAGMA)[9] through the Functional Mapping and Annotation of GWAS (FUMA)[10] pipeline to functionally annotate and biologically interpret the MTAG results. In total, ANNOVAR and MAGMA highlighted 1816 genes associated with CRP levels (Supplementary Data 6). Of those, 1245 genes were also associated with at least one of the other examined traits, and 23 of them were associated with all six traits possibly indicating a wide pleiotropic effect (Supplementary Data 7, Supplementary Fig. 4).

MAGMA gene-set analysis highlighted 19 CRP-associated gene-sets ($P < 3.2 \times 10^{-6}$) with various pleiotropic effects (Supplementary Data 8). In particular, the gene-set of nucleic acid binding was significantly associated with CRP, HDL, LDL, TG, and BMI. CRP-mapped genes were enriched for tissue expression showing higher expression in the liver ($P = 2.7 \times 10^{-4}$), and pituitary ($P = 1.3 \times 10^{-3}$) (Supplementary Data 9, Supplementary Fig. 5A), brain cerebellum ($P = 2.7 \times 10^{-5}$) and brain cerebellar hemisphere ($P = 1.1 \times 10^{-4}$) (Supplementary Data 10, Supplementary Fig. 5B).

### Colocalization of CRP loci and investigation of pleiotropy

We used HyPrColoc to investigate colocalization between the MTAG-reported 324 CRP loci and the five other traits examined in this study. We found 102 colocalized loci between CRP and at least one other trait

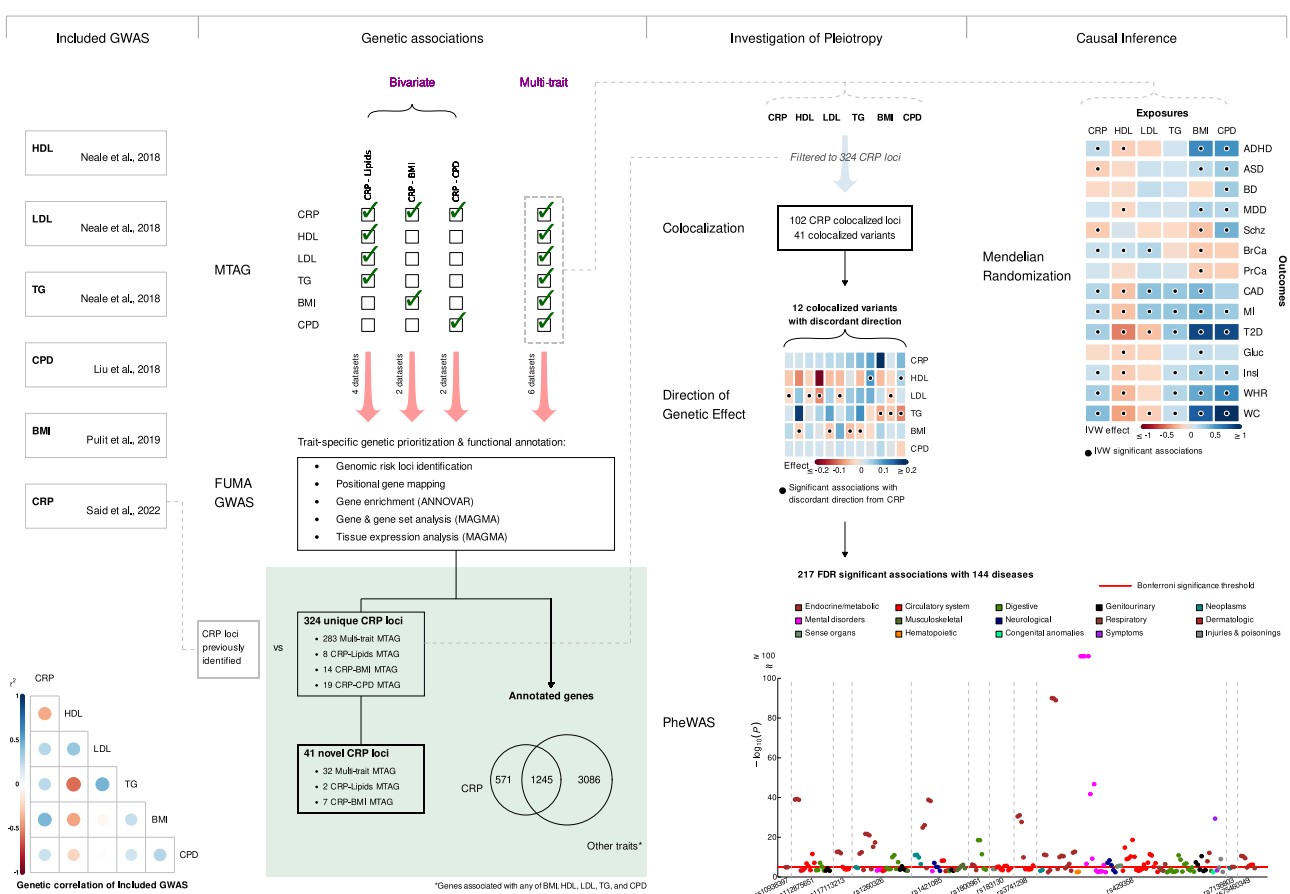

**Fig. 1 | Study design schematic for the discovery of novel CRP loci and investigation of pleiotropic loci.** BMI Body mass index, CPD Cigarettes per day, CRP C-reactive protein, HDL high-density lipoprotein, LDL low-density lipoprotein, TG triglycerides, ADHD attention deficit hyperactivity disorder, ASD autism spectrum

disorder, BD bipolar disease, MDD major depressive disorder, Schz schizophrenia, BrCa breast cancer, PrCa prostate cancer, CAD coronary artery disease, MI myocardial infarction, T2D type 2 diabetes, Gluc fasting glucose, Insl fasting insulin, WHR waist-to-hip ratio, WC waist circumference.

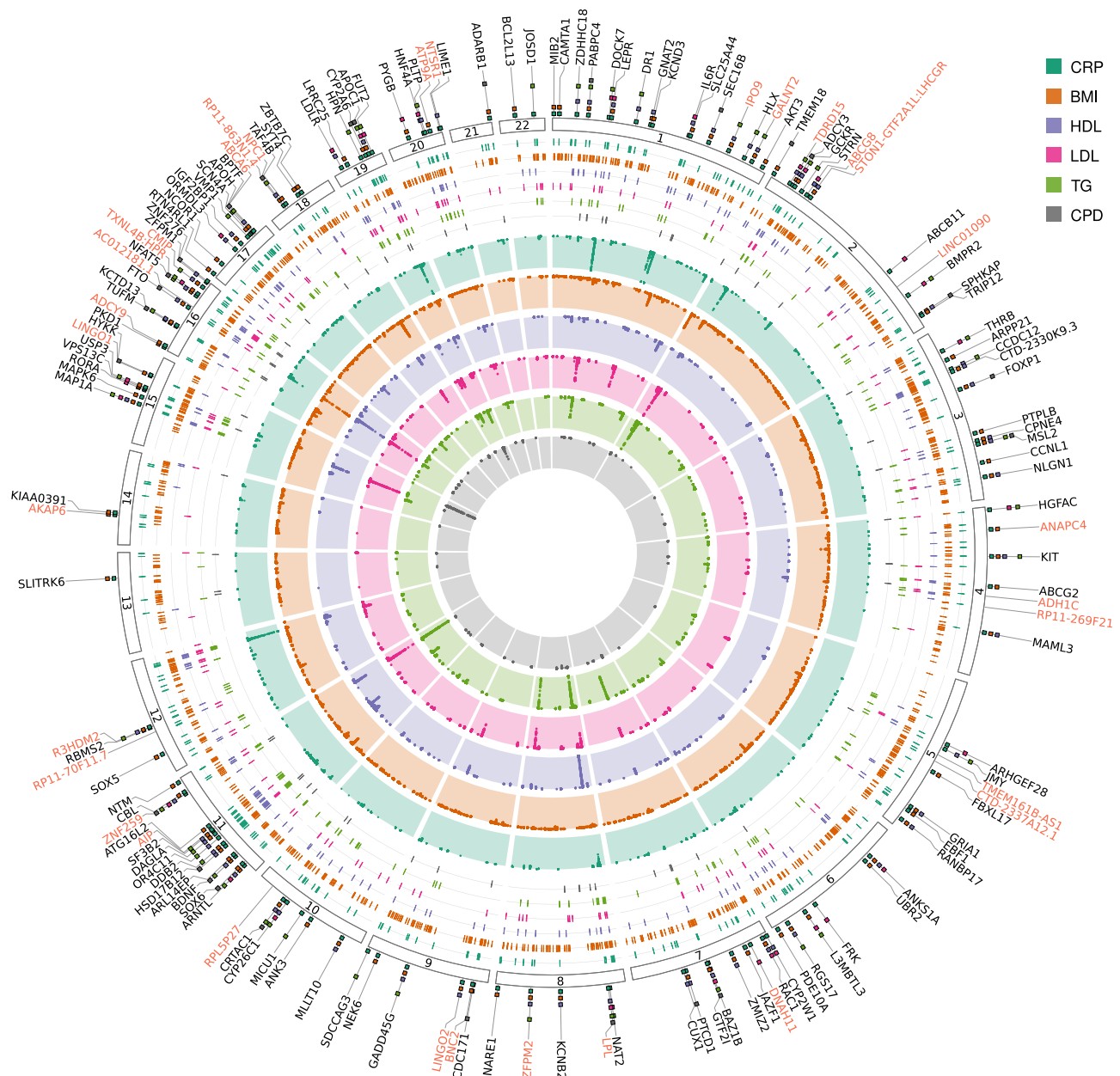

**Fig. 2 | Circular plot with MTAG results between C-reactive protein levels, high-density lipoprotein levels, low-density lipoprotein levels, triglyceride levels, body mass index and cigarettes per day.** The inner part of the graph presents the Manhattan plots of the genome-wide significant variants ($P < 5 \times 10^{-8}$) for each trait. The middle part presents the trait-specific genomic risk regions. The outer part presents the mapped genes of the CRP top signals which are either discovered as novel loci (red color) or additionally associated with any of the other examined traits. In the latter case, the colorful squares on the gene lines show the traits that the gene is associated with. The exact summary statistics are presented in Supplementary Data 1 and 3. BMI body mass index, CPD cigarettes per day, CRP C-reactive protein, HDL high-density lipoprotein, LDL low-density lipoprotein, TG triglycerides.

with a posterior probability (PP) > 0.8 (Supplementary Data 11). In 41 of those loci, we prioritized a single candidate causal variant for the colocalization (proportion of PP explained by variant ≥ 0.8) with 33 of them being CRP top signals in MTAG. Out of those 33 colocalized CRP top signals, 9 were also CRP novel signals (Table 2, Supplementary Figs. 6, 7). The exonic variant rs1260326 mapping at the *GCKR* gene colocalized with all six traits of interest (PP = 0.94) explaining 100% of the PP of the shared association. Also, an intergenic variant upstream (9 kb) from the *CRP* gene (rs2211320) was found to colocalize with CRP, HDL, LDL, TG, and CPD (PP = 0.97) with 99.95% of the shared association explained by this SNP. Novel CRP-associated variants rs34811474 (*ANAPC4*), rs10968576 (*LINGO2*), and rs879620 (*ADCY9*), colocalized with HDL, BMI, and CPD, respectively. Of the 41 colocalized

variants, 12 had a discordant direction of effects between the examined traits (MTAG $P < 0.05$) (Supplementary Data 12, Fig. 3). For those 12 variants, we further performed a phenome-wide association analysis (PheWAS)[11] in UK Biobank (UKB) to investigate possible discordant effects in clinical outcomes as well (Supplementary Data 13, Fig. 4). Of them, 7 variants (rs1260326, rs112875651 and rs1421085, rs117113213, rs1800961, rs183130, rs429358) had discordant FDR significant associations with various disease phenotypes. For example, rs1260326 (*GCKR*) which was associated with higher CRP and LDL levels, but lower BMI in MTAG, was also associated with 28 diseases in PheWAS including direct associations with disorders of lipid metabolism ($P = 2 \times 10^{-22}$), gout ($P = 3.9 \times 10^{-18}$), and angina pectoris ($P = 3.9 \times 10^{-4}$) and inverse associations with type 2 diabetes ($P = 3.6 \times 10^{-13}$),

**Table 1 | Novel genomic loci associated with C-reactive protein in genome-wide significance level (two-sided $P < 5 \times 10^{-8}$)**

| SNP | Chr:Pos | EA | OA | EAF | Beta | SE | P | Gene |
|---|---|---|---|---|---|---|---|---|
| rs545152 | 1:96886504 | T | C | 0.366 | 0.012 | 0.002 | $4.7 \times 10^{-8}$ | UBE2WP1 |
| rs6587552 | 1:151018861 | A | G | 0.229 | 0.013 | 0.002 | $2.5 \times 10^{-8}$ | BNIPL |
| rs8024 | 1:201845575 | A | C | 0.301 | 0.012 | 0.002 | $4.0 \times 10^{-8}$ | IPO9 |
| rs11122456 | 1:230305966 | A | G | 0.391 | 0.013 | 0.002 | $8.8 \times 10^{-10}$ | GALNT2 |
| rs754524 | 2:21311541 | G | T | 0.257 | 0.013 | 0.002 | $8.4 \times 10^{-9}$ | TDRD15:APOB |
| rs76866386 | 2:44075483 | C | T | 0.079 | −0.026 | 0.004 | $1.7 \times 10^{-10}$ | ABCG8 |
| rs4519576 | 2:48966146 | C | T | 0.450 | 0.011 | 0.002 | $4.2 \times 10^{-8}$ | STON1-GTF2A1L:LHCGR |
| rs12998606 | 2:188725859 | G | A | 0.461 | −0.011 | 0.002 | $2.3 \times 10^{-8}$ | LINC01090 |
| rs566279474 | 3:44135752 | C | T | 0.003 | 0.055 | 0.010 | $1.5 \times 10^{-8}$ | MIR138-1 |
| rs171390 | 3:154038412 | C | T | 0.424 | 0.011 | 0.002 | $4.7 \times 10^{-8}$ | DHX36 |
| rs2606227 | 3:183536836 | T | C | 0.361 | 0.012 | 0.002 | $4.5 \times 10^{-8}$ | MAP6D1 |
| rs34811474 | 4:25408838 | A | G | 0.217 | −0.015 | 0.002 | $6.3 \times 10^{-10}$ | ANAPC4 |
| rs1229978 | 4:100256199 | C | T | 0.406 | 0.012 | 0.002 | $1.7 \times 10^{-9}$ | ADH1C |
| rs1450786 | 4:112653076 | G | A | 0.367 | −0.011 | 0.002 | $3.7 \times 10^{-8}$ | RP11-269F21.1 |
| rs10461497 | 5:63942398 | C | T | 0.494 | −0.011 | 0.002 | $3.3 \times 10^{-8}$ | MRPL49P1 |
| rs6870983 | 5:87697533 | T | C | 0.236 | −0.014 | 0.002 | $7.3 \times 10^{-9}$ | TMEM161B-AS1 |
| rs11135450 | 5:95554016 | A | G | 0.345 | −0.012 | 0.002 | $2.4 \times 10^{-8}$ | CTD-2337A12.1 |
| rs2228213 | 6:12124855 | A | G | 0.333 | −0.012 | 0.002 | $1.8 \times 10^{-8}$ | HIVEP1 |
| rs2635727 | 6:50820940 | T | C | 0.250 | −0.014 | 0.002 | $2.5 \times 10^{-9}$ | RPS17P5 |
| rs57648913 | 7:21602065 | A | G | 0.145 | 0.016 | 0.003 | $2.4 \times 10^{-9}$ | DNAH11 |
| rs35237252 | 8:19870271 | A | C | 0.273 | −0.021 | 0.002 | $5.1 \times 10^{-20}$ | LPL |
| rs10464844 | 8:106419754 | G | A | 0.230 | 0.014 | 0.002 | $2.0 \times 10^{-8}$ | RP11-127H5.1:ZFPM2 |
| rs1411432 | 9:16728532 | C | A | 0.183 | 0.015 | 0.003 | $1.1 \times 10^{-8}$ | BNC2 |
| rs10968576 | 9:28414339 | G | A | 0.302 | 0.013 | 0.002 | $1.1 \times 10^{-9}$ | LINGO2 |
| rs722564 | 10:118550831 | T | C | 0.386 | −0.011 | 0.002 | $4.3 \times 10^{-8}$ | RPL5P27 |
| rs36089024 | 11:67244644 | T | C | 0.401 | −0.012 | 0.002 | $1.6 \times 10^{-9}$ | AIP |
| rs10750096 | 11:116656788 | C | A | 0.093 | 0.021 | 0.004 | $4.1 \times 10^{-8}$ | ZNF259 |
| rs7138803 | 12:50247468 | A | G | 0.338 | 0.012 | 0.002 | $1.5 \times 10^{-8}$ | RP11-70F11.7 |
| rs56205943 | 12:57679414 | A | G | 0.193 | −0.013 | 0.002 | $4.9 \times 10^{-8}$ | R3HDM2:RP11-123K3.4 |
| rs825457 | 12:124538302 | C | A | 0.161 | −0.016 | 0.003 | $2.7 \times 10^{-8}$ | FAM101A |
| rs17522122 | 14:33302882 | T | G | 0.489 | 0.012 | 0.002 | $2.6 \times 10^{-9}$ | AKAP6 |
| rs11856579 | 15:78012688 | A | G | 0.220 | −0.013 | 0.002 | $2.1 \times 10^{-8}$ | LINGO1 |
| rs879620 | 16:4015729 | C | T | 0.389 | −0.012 | 0.002 | $1.7 \times 10^{-9}$ | ADCY9 |
| rs12446515 | 16:56987015 | T | C | 0.292 | −0.020 | 0.002 | $1.3 \times 10^{-19}$ | AC012181.1 |
| rs2000999 | 16:72108093 | A | G | 0.192 | 0.015 | 0.003 | $9.8 \times 10^{-9}$ | TXNL4B:HPR |
| rs56823429 | 16:81533789 | C | A | 0.283 | 0.014 | 0.002 | $3.6 \times 10^{-10}$ | CMIP |
| rs77542162 | 17:67081278 | G | A | 0.011 | 0.048 | 0.007 | $5.4 \times 10^{-12}$ | ABCA6 |
| rs9951447 | 18:20009691 | C | T | 0.439 | −0.012 | 0.002 | $7.3 \times 10^{-9}$ | RP11-863N1.4 |
| rs2236707 | 18:21114997 | T | C | 0.430 | −0.014 | 0.002 | $1.0 \times 10^{-11}$ | NPC1 |
| rs2147338 | 20:50320079 | C | T | 0.406 | 0.011 | 0.002 | $2.8 \times 10^{-8}$ | ATP9A |
| rs3746778 | 20:61341472 | A | G | 0.392 | −0.011 | 0.002 | $3.0 \times 10^{-8}$ | NTSR1 |

*Chr:Pos* chromosome:position, *EA* effect allele, *OA* other allele, *EAF* effect allele frequency, *P* P-value, *Gene* mapped gene of the SNP.

cholelithiasis ($P = 9.5 \times 10^{-12}$), alcoholism ($P = 1.8 \times 10^{-4}$), and fasciitis ($P = 6.7 \times 10^{-4}$). Another variant, rs1421085 (*FTO*) was associated with 27 diseases including direct associations with obesity ($P = 5.6 \times 10^{-39}$), type 2 diabetes ($P = 8.2 \times 10^{-27}$), and hypertension ($P = 6.4 \times 10^{-9}$) and inverse with breast cancer ($P = 1 \times 10^{-11}$), and fasciitis ($P = 3.1 \times 10^{-4}$).

**Mendelian randomization**

We performed two-sample Mendelian randomization (MR) analysis to investigate the causality between CRP levels and a variety of prespecified outcomes with potential biological connection with CRP[12]. We used summary statistics generated from the multi-trait MTAG analysis and we further performed subgroup analysis keeping the CRP-independent SNPs without pleiotropic evidence and separately those that were pleiotropic based on colocalization. To allow comparison between CRP and the rest of the examined traits (HDL, LDL, TG, BMI, CPD), we extended the MR analysis to each one of them.

Genetically predicted higher CRP levels were associated with a lower risk of schizophrenia with consistent results across sensitivity analyses (Supplementary Data 14, Fig. 5) and breast cancer. The remaining outcomes examined did not show strong evidence for a causal association as sensitivity analyses were not supportive of the main IVW analysis suggesting pleiotropy (e.g., ischemic heart disease and diabetes). In subgroup analyses limited to the SNPs that were associated with CRP levels only, no outcome showed strong evidence for a causal effect except for schizophrenia which presented modest

**Table 2 | C-reactive protein top signals which colocalize (PP > 0.8) with any combination of the other examined traits explaining at least the 80% of the shared association**

| Causal SNP | Chr:Pos | Traits | PP | PP% | Gene |
|---|---|---|---|---|---|
| rs75460349 | 1:27180088 | CRP, TG | 1.00 | 80.1 | ZDHHC18 |
| rs61812598 | 1:154420087 | CRP, HDL, LDL | 0.96 | 88.3 | IL6R |
| rs2211320 | 1:159693605 | CRP, HDL, LDL, TG, CPD | 0.97 | 100.0 | CRP |
| rs4658403 | 1:243832560 | CRP, HDL, BMI | 0.95 | 99.7 | AKT3 |
| rs1260326 | 2:27730940 | CRP, HDL, LDL, TG, BMI, CPD | 0.94 | 100.0 | GCKR |
| rs17326656 | 2:48962291 | CRP, HDL, TG | 0.93 | 98.6 | STON1-GTF2A1L:LHCGR |
| rs2161037 | 2:169893419 | CRP, LDL | 1.00 | 99.9 | ABCB11 |
| rs6792725 | 3:24520283 | CRP, LDL, TG | 1.00 | 100.0 | THRB |
| rs171390 | 3:154038412 | CRP, BMI | 0.98 | 87.0 | DHX36 |
| rs247975 | 3:173107443 | CRP, HDL, TG, BMI, CPD | 0.80 | 88.9 | NLGN1 |
| rs34811474 | 4:25408838 | CRP, HDL, BMI, CPD | 0.95 | 100.0 | ANAPC4 |
| rs10938397 | 4:45182527 | CRP, HDL, BMI | 0.86 | 82.7 | RP11-362I1.1 |
| rs6870983 | 5:87697533 | CRP, BMI | 1.00 | 100.0 | TMEM161B-AS1 |
| rs2228213 | 6:12124855 | CRP, BMI | 0.96 | 90.6 | HIVEP1 |
| rs5017416 | 6:18492350 | CRP, BMI | 1.00 | 94.0 | RNF144B |
| rs1490384 | 6:126851160 | CRP, LDL | 0.81 | 92.5 | MIR588 |
| rs35237252 | 8:19870271 | CRP, CPD | 0.98 | 92.8 | LPL |
| rs112875651 | 8:126506694 | CRP, LDL, TG | 1.00 | 100.0 | RP11-136O12.2 |
| rs7031064 | 9:14455076 | CRP, BMI | 1.00 | 99.4 | NFIB |
| rs10968576 | 9:28414339 | CRP, HDL, BMI, CPD | 0.87 | 85.6 | LINGO2 |
| rs11012732 | 10:21830104 | CRP, BMI | 0.99 | 85.6 | MLLT10 |
| rs6486122 | 11:13361524 | CRP, HDL, LDL, TG | 0.99 | 96.1 | ARNTL |
| rs6265 | 11:27679916 | CRP, BMI, CPD | 0.93 | 91.6 | BDNF-AS:BDNF |
| rs4755720 | 11:43628749 | CRP, HDL, TG | 0.96 | 80.2 | HSD17B12 |
| rs3741298 | 11:116657561 | CRP, LDL, TG | 0.95 | 100.0 | ZNF259 |
| rs7138803 | 12:50247468 | CRP, HDL, BMI | 0.99 | 100.0 | RP11-70F11.7 |
| rs9604045 | 13:113927208 | CRP, HDL | 0.99 | 100.0 | LDHBP1 |
| rs2239222 | 14:73011885 | CRP, LDL | 1.00 | 100.0 | RGS6 |
| rs11635675 | 15:63793238 | CRP, LDL, TG, BMI | 0.94 | 81.3 | USP3 |
| rs11852372 | 15:78801394 | CRP, TG, CPD | 0.83 | 99.7 | HYKK |
| rs879620 | 16:4015729 | CRP, HDL, BMI, CPD | 0.98 | 100.0 | ADCY9 |
| rs3814883 | 16:29994922 | CRP, HDL, TG, BMI | 0.93 | 100.0 | TAOK2 |
| rs1421085 | 16:53800954 | CRP, HDL, LDL, BMI, CPD | 0.98 | 100.0 | FTO |
| rs183130 | 16:56991363 | CRP, HDL, LDL, TG, CPD | 0.89 | 100.0 | AC012181.1 |
| rs2000999 | 16:72108093 | CRP, LDL, TG | 0.99 | 88.1 | TXNL4B:HPR |
| rs2925979 | 16:81534790 | CRP, HDL, TG | 0.93 | 100.0 | CMIP |
| rs56113850 | 19:41353107 | CRP, BMI, CPD | 0.95 | 100.0 | CTC-490E21.12:CYP2A6 |
| rs429358 | 19:45411941 | CRP, BMI, CPD | 1.00 | 100.0 | APOE |
| rs117113213 | 20:39165692 | CRP, LDL, TG | 0.99 | 99.4 | SNORD112 |
| rs1800961 | 20:43042364 | CRP, HDL, LDL | 1.00 | 100.0 | HNF4A |
| rs397092 | 21:46582564 | CRP, HDL, TG, BMI, CPD | 0.91 | 96.0 | ADARB1 |

*Causal SNP* candidate variant causing the colocalization, *Chr:Pos* Chromosome:Position, *Traits* traits which colocalize, *PP* posterior probability, *PP* % percentage of PP explained by SNP, *Gene* mapped gene, *BMI* body mass index, *CPD* cigarettes per day, *CRP* C-reactive protein, *HDL* high-density lipoprotein, *LDL* low-density lipoprotein, *TG* triglyceride.

evidence for inverse associations with genetically determined higher CRP levels (Supplementary Data 14).

## Discussion

Using multi-trait analysis of GWAS between CRP and five cardiometabolic risk factors we boost the statistical power to search for genomic variants associated with circulating CRP levels identifying 41 novels CRP loci. Moreover, we present a comprehensive overview of the pleiotropic genetic architecture of this trait and indicate 19 gene sets associated with CRP as potential master regulators of chronic low-grade inflammation with wide pleiotropic effects on lipids and other cardiometabolic pathways. Through subsequent colocalization analysis, we identified 41 shared causal variants between CRP and cardiometabolic risk factors and further examine the associations across the phenome for 12 colocalized variants with discordant effect direction between the traits. MR analysis provided evidence for a causal effect of low-grade chronic inflammation as measured by genetically predicted serum CRP on a lower risk of schizophrenia supporting findings from the previous studies[12]. Evidence for causality on other diseases was limited when pleiotropic signals were excluded.

The multi-trait GWAS via MTAG leveraged the correlation between C-reactive protein with its major determinants (lipids, BMI,

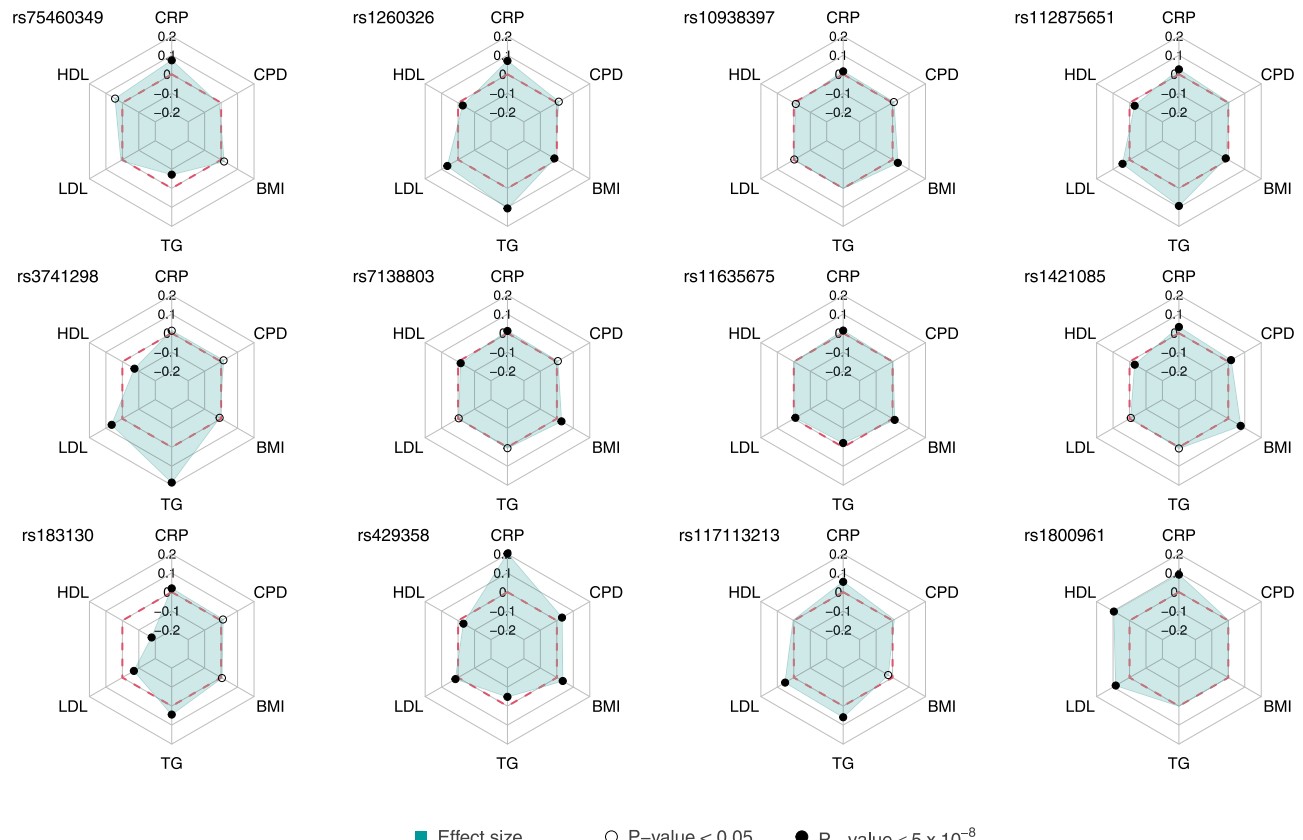

**Fig. 3 | Effect size across traits for the 12 colocalized SNPs with a discordant direction of effect between C-reactive protein levels and any of the other examined traits.** Inverse effect directions between CRP and any of LDL, TG, BMI, and CPD or direct directions between CRP and HDL were considered discordant if the association of the SNP with the discordant trait was statistically significant ($P < 0.05$) in multi-trait MTAG. The red dashed line represents the zero value of the effect. Significant associations ($P < 0.05$) are presented with white-filled circles while genome-wide significant (GWS) associations are with black-filled circles. The exact summary statistics are provided in Supplementary Data 12. *BMI* body mass index, CPD cigarettes per day, CRP C-reactive protein, *HDL* high-density lipoprotein, *LDL* low-density lipoprotein, *TG* triglycerides.

and cigarette smoking) to increase the number of identified loci and the informativeness of the bioinformatics analysis. Our findings support the close link between CRP and lipid metabolism. Many of the novel CRP loci were well-known lipid and BMI loci such as variants mapped at *LPL*, *APOB*, and *LINGO2* genes which had not been associated with CRP previously in univariate GWAS analyses. At the same time, several novel CRP loci have not been previously associated with any of the other examined traits highlighting the additional power of MTAG analysis[13]. Prioritized CRP loci were enriched for expression not only in the liver and pituitary but also in the brain cerebellum and cerebellar hemisphere underlining the importance of those pleiotropic loci in this tissue.

An exonic variant (rs2228213) within the human immunodeficiency virus type I enhancer-binding protein 1 (*HIVEP1*) gene was found to colocalize between CRP and BMI and was highlighted as a novel locus for CRP from MTAG. HIVEP1 regulates the transcription of inflammatory target genes such as those belonging to the interleukin signaling pathway and HIVEP1 deficiency has been shown to exacerbate inflammation in sepsis[14]. The colocalized exonic variant provides a possible mechanism for the interaction between infection, inflammation, and adiposity.

A non-synonymous exon variant (rs34811474) within the anaphase-promoting complex subunit 4 (*ANAPC4*) gene was identified as the causal variant for multiple traits including CRP, HDL, BMI, and CPD. Whereas high pleiotropy at the locus and gene level is common across the genome, wide pleiotropic effects at the SNP level are much less abundant[15]. The same variant has been previously associated with cognitive performance and educational attainment[16], lung function[17],

and osteoarthritis[18]. The biological function of ANAPC4 is largely unknown but it's wide pleiotropic effects and its low tissue specificity indicate a possible involvement in general biological functions.

The *DHX36* gene (rs171390) was found to colocalize between CRP and BMI and was highlighted as a novel locus for CRP from MTAG. DHX36, a highly conserved member of the DExD/H box helicase family, binds with and unfolds G-Quadruplex (G4) DNA, thereby changing how G4 structures influence DNA- and RNA-dependent processes[19]. G4 structures and DHX36 interactivity has been studied in relation to cancer and tumorigenesis, neurodegenerative diseases and aging mechanisms including cellular senescence[19]. Our findings support a possible role of this gene in these traits through inflammation pathways and adiposity.

Other highlighted variants which colocalized across multiple traits include variants within well-studied genes such as the glucokinase regulatory protein (*GCKR*). The common non-synonymous mutation of rs1260326 close to the *GCKR* gene is widely reported to have pleiotropic effects on cardiometabolic traits and has been associated with increased triglyceride levels and decreased glucose levels[20]. Our study supported these observations. In particular, a discordant direction of effect across CRP, lipids, and BMI was observed which was also translated into a discordant direction of effects with 28 different disease phenotypes.

Our study had several strengths. First, we used large-scale data with substantial sample sizes of over 500,000 participants each resulting in high statistical power. The statistical power of our study was boosted even higher considering that we analyzed the data by performing suitable multivariate methods. Second, a particular

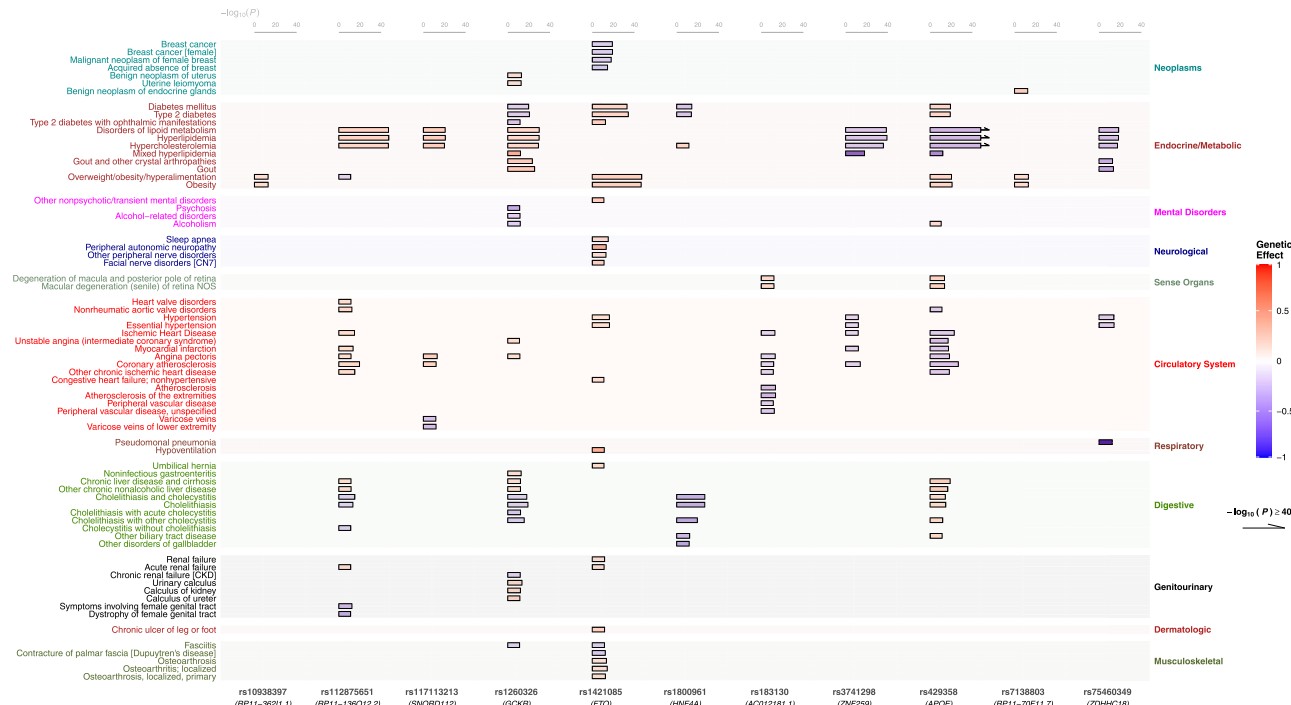

**Fig. 4 | Highlighted PheWAS results (FDR significant) in UK Biobank for the 12 colocalized SNPs with a discordant direction of effect between C-reactive protein levels and any of the other examined traits.** The color of each bar represents the genetic effect size for the respective SNP while the length of each bar shows the statistical significance on the $-\log_{10}P$ scale. The variant rs429358 (*APOE*) presented 96 FDR significant associations that can be found in Supplementary Data 13. To optimize the visualization of the results, the diseases that were associated only with that variant were removed from the graph. The variant rs11635675 (*USP3*) presented no FDR significant association and thus it was removed from the graph. The exact summary statistics are provided in Supplementary Data 13.

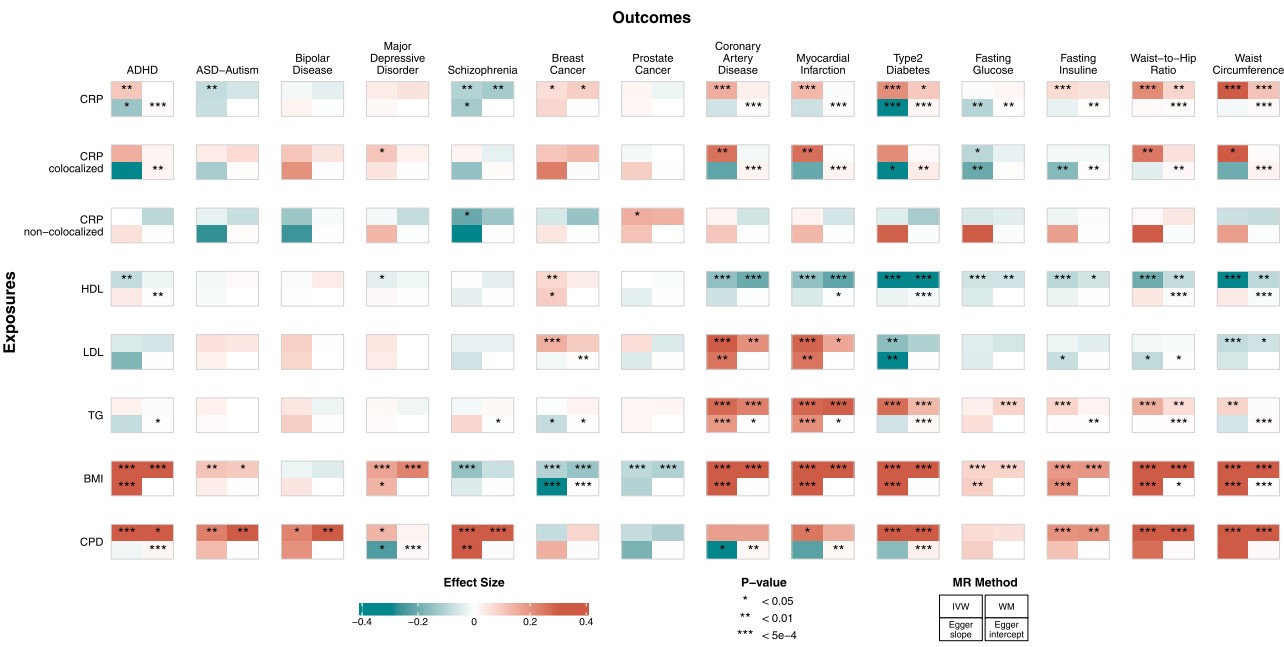

**Fig. 5 | Two-sample Mendelian randomization results.** Each row is a different exposure and each column is a different outcome. Each Mendelian randomization analysis is summarized in a four-squared box including the estimates from inverse-variance weighted method (IVW, top left square), weighted Median (WM, top right), Egger method (slope, bottom left; intercept, bottom right). The color of each square represents the respective effect size while the asterisks indicate the statistical significance. The exact summary statistics are provided in Supplementary Data 14. BMI body mass index, CPD cigarettes per day, CRP C-reactive protein, HDL high-density lipoprotein, LDL low-density lipoprotein, TG triglycerides, ADHD attention deficit hyperactivity disorder, ASD autism spectrum disorder.

strength of our study is that it combines advanced methods of genetic epidemiology such as multivariate analysis, colocalization, PheWAS, and Mendelian randomization. Third, we investigated possible associations between CRP and many risk factors for various complex diseases including CVDs, mental disorders, and neoplasms as well as other diseases. However, our study also had some limitations. Our analysis was restricted to individuals of European ancestry only and results may not be generalizable in other ancestries. Also, we excluded all rare variants (MAF < 1%) from our MTAG analysis and consequently from all following analyses. Therefore, we may not be able to identify rare variants with large effects, However, we should consider that several of these associations may stem from false-positive findings and biased results. Moreover, MTAG was conducted using summary statistics data and hence, it was impossible to adjust for other confounders. We should also pinpoint, there was sample overlap between the studies in the multi-trait analysis. However, the sample overlap was addressed within each GWA study by using bivariate linkage disequilibrium (LD) score regression[13]. Fifth, we defined pleiotropy as the presence of statistically significant associations between a genetic variant and more than one phenotype. Our definition, therefore, refers to 'statistical pleiotropy' and includes situations of horizontal pleiotropy (for example, one SNP directly influences multiple phenotypes), and situations of vertical pleiotropy where statistical associations to multiple traits are induced via causal effects of one trait on another or via a third common factor[15]. Although detecting shared genetic associations between two or more traits suggest that possibly there are pleiotropic genes involved in the biological pathways of all the associated traits, still, this does not necessarily mean that the traits share biological pathways, as the pleiotropic genes could affect the traits independently via different pathways, or they could even be expressed in different tissues in response to different signaling[21,22]. The extensive pleiotropy at the CPR loci is likely to bias the MR assumption of no pleiotropy. However, the analyses, limiting genetically predicted CRP to variants that were not also associated with lipids or BMI resulted in a small number of CRP variants and suffering from small power. Furthermore, our GWAS introduced several novel genes for CRP, but high polygenicity and pleiotropy of the trait pose a challenge to uncovering biological mechanisms, especially as many novel variants each have a small effect on CRP. Finally, larger GWAS on lipids or CRP could increase the number of pleiotropic loci in the future and further expand our knowledge of the link between mechanisms of inflammation and cardiometabolism.

In conclusion, we performed a comprehensive multi-trait analysis on CRP and cardiometabolic traits discovering 41 novels of CRP genetic loci. Our subsequent colocalization analysis further highlighted 41 shared causal variants between inflammation and cardiometabolism implying that a perturbation in these loci might be expected to affect several of these traits. Our comprehensive analysis of pleiotropic effects, therefore, offers a promising path forward for novel preventive and therapeutic targets and for the study of anticipated side effects across several traits. Functional work to elucidate causal variants and mechanisms at these loci may provide further insights into the etiological pathways for this collection of traits.

## Methods

### Study design and population samples

The summary statistics from six GWAS were used in this study: CRP, HDL, LDL, TG, BMI, and CPD. As a rule, the largest GWAS of European ancestry for the respective trait was used. All GWAS used in this study has exclusively or partially been conducted in UK Biobank (UKB). More specifically, the CRP summary statistics were obtained from the GWAS meta-analysis of UKB and CHARGE-1000 Genomes[12]; HDL, LDL, and TG from the respective GWAS in UKB (UK Biobank−Neale lab)[23]; BMI from the GWAS meta-analysis of UKB and GIANT[24]; and CPD from a meta-analysis of over 30 GWAS including UKB[25].

Although all the above GWAS have been described elsewhere, a brief presentation of each GWAS characteristic can be found in Supplementary Data 15.

### Genotypic quality control

The initial datasets contained 11,140,987; 13,791,468; 13,791,468; 13,791,468; 27,381,303, and 12,003,614 genotyped and imputed SNPs in CRP, lipids (HDL, LDL, TG), BMI and CPD, respectively. All insertion and deletion polymorphisms, rare variants (MAF < 0.01), variants with a sample size <2/3 of the 90th percentile and palindromic SNPs were excluded from the analysis. Furthermore, all non-overlapping SNPs that were not present in at least one dataset were further excluded from the analysis leaving 6,206,408 SNPs for analysis.

### Multi-trait GWAS analysis

A multi-trait analysis of GWAS (MTAG)[13] was performed to jointly analyze the summary statistics of CRP, HDL, LDL, TG, BMI, and CPD. The genetic correlation between the traits was calculated and further corrected for sample overlap using bivariate LD score regression which estimated the correlation in GWAS estimation error as implemented in MTAG. Distinct trait-specific effect estimates were generated for each SNP resulting in six summary statistic datasets in total, one for each trait. Thus, a distinct P-value per SNP was generated for each trait which can be interpreted like those from a univariate single-trait GWAS.

As well as analyzing the six traits jointly, three additional bivariate MTAG analyses were performed: CRP with lipids; CRP with BMI; and CRP with CPD, using the same methodology as described above.

### Functional mapping and annotation

The generated summary results from all the multivariate GWAS were further functionally analyzed using the platform functional mapping and annotation of GWAS (FUMA v.1.3.6a)[10] in a total of 14 distinct functional analyses (6 from the multi-trait MTAG; 4 from the CRP-lipids MTAG; 2 from the CRP-BMI MTAG; and 2 from the CRP-CPD MTAG). In each one of them, a similar process was implemented which is described as follows. All trait-specific genome-wide significant SNPs at a predefined threshold of $P = 5 \times 10^{-8}$ were identified and clumped two times at a different $r^2$ threshold each time. The first clumping at $r^2 < 0.6$ was used to determine the coordinates of the genomic risk loci. All the SNPs that survived the first clumping were included for further annotation and gene prioritization. A second clumping was performed afterward at $r^2 < 0.1$ to define independent signals. SNPs in LD with each other at $0.1 \le r^2 < 0.6$ were assigned to the same LD block. The LD blocks were merged into one locus if there were SNPs from different LD blocks closer than 500 kb. The SNPs that survived the second clumping were defined as independent SNPs. Independent SNPs with the smallest P-value in each region were defined as the top lead SNPs representing the corresponding genomic risk loci. The European sample of 1000 Genome Project Phase 3 was used as a reference panel to calculate pairwise LD between SNPs using PLINK v1.9[26]. SNPs were positionally mapped to their nearest protein-coding genes (Ensembl build v92) at a maximum distance of 10 kb using ANNOVAR[8].

### CRP novel genetic loci definition

As CRP was analyzed in four distinct MTAG analyses (1 multi-trait and 3 bivariate), we gathered all CRP-independent signals from all four distinct CRP-generated summaries and after excluding the same signals or proxies (either in distance ±500 kb or in LD $r^2 > 0.1$), we compared our top CRP independent signals with those from the UKB-CHARGE GWAS meta-analysis on CRP[12]. The top CRP signals from our analyses in absolute distance >500 kb and $r^2 < 0.1$ from all previously reported independent CRP signals were considered novel signals and their region's novel loci.

## Gene-based, gene-set, and gene-property analysis: Tissue expression analysis

We used Multi-marker Analysis of GenoMic Annotation software (MAGMA v1.08)[9] to perform gene, gene-set, and gene-property analysis. The gene analysis was performed for SNPs physically located in a gene using the reference panel of 1000 Genomes phase 3. An SNP-wide mean model for gene tests was implemented following the default parameters of the FUMA pipeline. In the gene-set analysis, we tested 15,477 different gene-sets derived from MsigDB v7.0[27]. The results from both gene and gene-set analyses were corrected for multiple tests implementing the Bonferroni correction (*P*-value threshold: $2.7 \times 10^{-6}$ and $3.2 \times 10^{-6}$, respectively).

The gene-property analysis was performed to test possible associations between tissue-specific gene expression profiles and the genes found to be associated with the trait of interest. The gene expression data sets were obtained from the Genotype-Tissue Expression version 8 (GTEx v8) testing a positive relationship at a Bonferroni significance level between genetic associations and gene expression in 30 general and 53 more specific tissue types, respectively.

## Colocalization

Multi-trait colocalization analysis was performed in R v.4.1.0[28] using HyPrColoc v.1.0.0R package[29] which is a Bayesian divisive clustering algorithm for identifying shared genetic associations between traits in a genomic region using GWAS summary statistics. More specifically, this method was performed to identify colocalized loci between CRP and any combination of the other examined traits and candidate distinct causal variants explaining the shared association in genomic regions associated with CRP.

We performed a distinct colocalization analysis for each CRP-associated top signal from MTAG in a region ±200 kb from the top SNP using the MTAG summary statistics. We considered variant-specific priors for our analyses, which assumes that the probability of a variant being colocalized with a set of traits decreases as the number of the set of traits increases. The variant-specific priors model requires the specification of two priors. We specified the prior probability that a variant is associated with a single trait only at $P = 1 \times 10^{-4}$ and a conditional prior probability that a variant is associated with an additional trait given that it is already associated with another trait at $P_c = 0.02$. A PP higher than 0.8 was considered strong evidence that the traits colocalize in the region. The variant with the highest proportion of PP explained was considered the candidate causal variant for the shared association if that proportion was at least 80%.

## Direction of genetic effect investigation and Phenome-Wide Association Analysis (PheWAS)

Variants found to colocalize with CRP and any combination of the other examined traits were further investigated focusing on the direction of their genetic effect across traits. For each distinct colocalized SNP, we compared the direction of its genetic effect on all examined traits using the summaries from the multivariate MTAG. Hence, we were able to highlight colocalized SNPs with a discordant direction of effect between CRP and any of the other traits. Opposite directions of effect between CRP and any of LDL, TG, BMI, and CPD or the same directions between CRP and HDL were considered discordant if the association with the discordant trait was statistically significant (*P* < 0.05).

Afterward, to investigate the impact of the colocalized genetic loci with the discordant direction of effect on the phenome, we performed a distinct PheWAS analysis in UKB for each one of those SNPs. The PheWAS analysis was restricted to participants of European ancestry and to ensure the independence of participants, we randomly excluded one participant from each pair of relatives based on their kinship coefficient > 0.0884 as provided by UKB. After the quality control

process, we ended up with 424,439 individuals available for the analysis.

We used the inpatient Hospital Episode Statistics (HES) records, cancer, and death registries, which follow the WHO's International Classification of Diseases coding system, 9th Revision (ICD-9) or 10th Revision (ICD-10). We used both ICD-9 and ICD-10 to define cases and controls after translating them to the phecode grouping system as implemented in PheWAS R package[30]. A series of case-control groups were generated for each phecode, with controls identified as individuals with no record of the respective outcome and its related phecodes.

To ensure adequate statistical power in these analyses, we tested only phecodes with a sample size ≥200 cases as previously proposed from simulation studies[31]. We used logistic regression models adjusting for age, sex, and the first 15 genetic principal components. To reduce false-positive signals due to multiple testing, we implemented the false discovery rate (FDR) method[32].

## Mendelian randomization

We used MR analysis to investigate causality between CRP and pre-specified outcomes which have previously shown associations with CRP and have been extensively studied. We performed multiple MR analyses, one for each combination between 6 exposures (CRP, HDL, LDL, TG, BMI, and CPD) and 14 outcomes including diseases of the circulatory system, neoplasms, mental disorders and metabolomic traits (Supplementary Data 16). For each exposure, the genetic instrumental variables were the independent SNPs (LD $r^2 < 0.1$, $P < 5 \times 10^{-8}$) of the respective trait from multivariate MTAG.

In all MR, our main analysis was the inverse-variance weighted (IVW) method assuming a random-effects model[33]. Additionally, two sensitivity analyses were performed: (1) the weighted median (WM) method to check if at least 50% of our SNPs were valid instruments[34] and (2) the MR-Egger method to test and correct for possible directional pleiotropy[35].

We performed a subgroup MR analysis on SNPs associated with CRP levels based on the existence or absence of evidence of pleiotropy. Two groups of SNPs were created; one with the colocalized CRP SNPs only and the other with non-colocalized SNPs that were non-significant (MTAG *P*-value > 0.01) for all the other examined traits.

## Reporting summary

Further information on research design is available in the Nature Portfolio Reporting Summary linked to this article.

# Data availability

The summary statistics of all GWAS used in this study are publicly available from GWAS Catalog (CRP: GCST90029070; BMI: GCST009004; CPD: GCST007459) or Neale lab (lipids: http://www.nealelab.is/uk-biobank). The MTAG summary statistics generated in this study have been deposited in NHGRI-EBI GWAS Catalog under accession codes GCST90179146 (CRP), GCST90179147 (HDL), GCST90179148 (LDL), GCST90179149 (TG), GCST90179150 (BMI) and GCST90179151 (CPD). All other data generated in this study are provided with this published article (and its supplementary information files).

# Code availability

No previously unreported custom computer code or mathematical algorithm was used to generate results central to the conclusions.

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

## Acknowledgements

This work was supported by the UK Dementia Research Institute at Imperial College, which receives its funding from UK DRI Ltd. (funded by the UK Medical Research Council, Alzheimer's Society, and Alzheimer's Research UK) and the British Heart Foundation Centre for Research Excellence at Imperial College London and the National Institute for Health Research Imperial Biomedical Research Centre, Imperial College London. This research work was supported by the Hellenic Foundation for Research and Innovation (H.F.R.I.) under the "First Call for H.F.R.I. Research Projects to support Faculty members and Researchers and the procurement of high-cost research equipment grant" (Project Number: HFRI-FM17-1312). The funding organizations had no role in the design and conduct of the study, including data collection through to interpretation and paper preparation, review, or approval.

## Author contributions

F.K., A.D., and I.T. designed the research, F.K. conducted the analyses, visualised the results and wrote the manuscript. F.K., E.E., A.D., and I.T. interpreted the results. E.E., S.S., J.J.B., P.E., A.D., and I.T. critically revised the manuscript.

## Competing interests

The authors declare no competing interests.
