## [Peer Review File · Nature Communications]

Pleiotropic genetic architecture and novel loci for C-reactive protein levelsREVIEWER COMMENTS

Reviewer #1 (Remarks to the Author):

Koskeridis et al. conducted a cross-trait genetic analysis to understand the pleiotropic genetic architecture and novel loci for circulating CRP levels. The authors found 41 novel CRP loci and 19 CRP associated gene sets with various pleiotropic effects. Additionally, 41 variants colocalized between CRP and cardiometabolic risk factors and 12 of them displayed unexpected discordant effects between the shared traits which were further translated into discordant associations with clinical outcomes in a subsequent PheWAS. The findings progress the previous knowledge with many novel CRP loci and provide insights into shared mechanisms underlying between inflammation and lipid metabolism which potentially represent novel preventive and therapeutic targets.

While the paper is technically sound, my biggest concern is its rational, as I feel the introduction fails to provide a comprehensive picture on why such an analysis has to be conducted with a special focus on cardiovascular traits (and the five selected components). As the authors mentioned in the discussion (also based on my own knowledge), CRP levels have been associated with a plethora of phenotypes and diseases including a number of (primarily) immune disorders and inflammations, why these diseases were not prioritized while cardiovascular diseases were instead investigated here? Have the immune disorders or inflammations with CRP already been studied using such a design? The introduction lacks an adequate description regarding the rational and the current progress of the field. How much are known and how much are unknown regarding the genetics, the pleiotropic architecture of CRP, should be briefly and clearly described in the intro, which is the standpoint for all subsequent analysis. Otherwise, such analytical schematics can be applied to any pair of complex diseases Furthermore, the results were a bit scattered without good interconnections. For SNPs with discordant effects, the authors performed a PheWas, however, in turns of MR analysis, a candidate approach was used where the numbers of outcomes were limited to less than 20. I don't quite get the underlying logics.

Reviewer #2 (Remarks to the Author):

The author conducted a multivariate analysis of CRP with a few other metabolic related traits, including lipids levels. The identified 41 novel loci and used colocalization analysis to highly variants with pleiotropic effects. Overall the study is interesting, but does not seem to be particularly novel and the improvement is rather incremental.

I have some major and a few minor concerns:

1) the used datasets do not represent the largest GWAS. For example, the authors used UKB lipids GWAS, while there are much bigger studies conducted by GLGC that already integrates UKB. This sounds like a missed opportunity to make more powerful discovery.

2) The term "pleiotropy" has numerous meanings. The meaning used in the article is not entirely clear. It seems to me that the pleiotropy here means the same "causal" variant associated with multiple traits (based on colocalization analysis). The mechanism of the association should be better illustrated. Is the observed multi-trait association induced by the causal relationship between traits, or due to the presence of a confounder that is simultaneously correlated with both traits, etc. Using a method such as Stephens 2012 PLOS ONE can be more informative for this purpose.

Reviewer #1 (Remarks to the Author):

Koskeridis et al. conducted a cross-trait genetic analysis to understand the pleiotropic genetic architecture and novel loci for circulating CRP levels. The authors found 41 novel CRP loci and 19 CRP associated gene sets with various pleiotropic effects. Additionally, 41 variants colocalized between CRP and cardiometabolic risk factors and 12 of them displayed unexpected discordant effects between the shared traits which were further translated into discordant associations with clinical outcomes in a subsequent PheWAS. The findings progress the previous knowledge with many novel CRP loci and provide insights into shared mechanisms underlying between inflammation and lipid metabolism which potentially represent novel preventive and therapeutic targets.

While the paper is technically sound, my biggest concern is its rationale, as I feel the introduction fails to provide a comprehensive picture on why such an analysis has to be conducted with a special focus on cardiovascular traits (and the five selected components). As the authors mentioned in the discussion (also based on my own knowledge), CRP levels have been associated with a plethora of phenotypes and diseases including a number of (primarily) immune disorders and inflammations, why these diseases were not prioritized while cardiovascular diseases were instead investigated here? Have the immune disorders or inflammations with CRP already been studied using such a design? The introduction lacks an adequate description regarding the rationale and the current progress of the field. How much are known and how much are unknown regarding the genetics, the pleiotropic architecture of CRP, should be briefly and clearly described in the intro, which is the standpoint for all subsequent analysis. Otherwise, such analytical schematics can be applied to any pair of complex diseases.

Reply: Thank you for positive feedback. While inflammation and CRP are linked to a wide range of diseases, we agree that our manuscript is focused on cardiometabolic traits. We therefore extensively reviewed the introduction to discuss the link between metabolism and immune systems and justify the rationale of our study. For example, we now state (p.1, lines 2-8) “Metabolic and immune are evolutionarily conserved and functionally entangled systems whose interplay is believed to play a central role in the pathophysiology of a wide spectrum of cardiovascular diseases¹. As a result, metabolic disturbances are commonly associated with immune responses in relation to cardiovascular outcomes. For example, high low-density lipoprotein (LDL) cholesterol levels and its subsequent modification are involved in the chronic inflammatory response of the vascular wall².” and (p.1, lines 19-21) “A deeper understanding of the interplay between inflammation and cardiovascular risk factors is likely to highlight important disease pathways and interventions with added clinical benefit.”

Furthermore, the results were a bit scattered without good interconnections. For SNPs with discordant effects, the authors performed a PheWAS, however, in turns of MR analysis, a candidate approach was used where the numbers of outcomes were limited to less than 20. I don't quite get the underlying logics.

Reply: Thank you for your comment. PheWAS was limited to single SNPs which had shown discordant effects across the phenotypes of interest. We applied PheWAS to examine whether this discordance was translated into different associations across the phenome. On the other hand, Mendelian Randomization was applied across the CRP MTAG discoveries in relation to a prespecified list of traits that have been previously reported from other studies to be associated with CRP (from the most

recent CRP GWAS; Said et al. Nat Commun. 2022). Our aim was to examine the impact of the novel loci on the previously reported Mendelian Randomization associations as well as whether those associations would be different if pleiotropic loci were excluded from the analyses. We now further explain the rationale of these analyses and have improved the connectivity of different results.

In results, we mention (p.4-5, lines 47-49) *“For those 12 variants, we further performed a phenome-wide association analysis (PheWAS)¹¹ in UK Biobank (UKB) to investigate possible discordant effects in clinical outcomes as well”* and (p.5, lines 61-63) *“We performed two-sample Mendelian randomization (MR) analysis to investigate the causality between CRP and a variety of prespecified outcomes with potential biological connection with CRP¹².”*

In methods, we mention (p.16, lines 101-103) *“Afterwards, to investigate the impact of the colocalized genetic loci with discordant direction of effect on the phenome, we performed a distinct PheWAS analysis in UKB for each one of those SNPs.”* and (p.17, lines 120-121) *“We used MR analysis to investigate causality between CRP and prespecified outcomes which have previously shown associations with CRP and have been extensively studied.”*

Reviewer #2 (Remarks to the Author):

The author conducted a multivariate analysis of CRP with a few other metabolic related traits, including lipids levels. The identified 41 novel loci and used colocalization analysis to highly variants with pleiotropic effects. Overall the study is interesting, but does not seem to be particularly novel and the improvement is rather incremental.

I have some major and a few minor concerns:

1) the used datasets do not represent the largest GWAS. For example, the authors used UKB lipids GWAS, while there are much bigger studies conducted by GLGC that already integrates UKB. This sounds like a missed opportunity to make more powerful discovery.

Reply: The GWAS datasets used in this multi-phenotype GWAS analysis were those which were publicly available at the time of the analysis. The reviewer is correct that larger GWAS has since been released but the summary datasets were not publicly available at the time we performed our multi-trait analyses. However, our analyses were based on large scale GWAS studies with large sample sizes and we believe that the three GWAS on lipids from UKB with a sample size of half a million people provide adequate statistical power for robust findings presented here. We added the following sentence to discussion (p.10, lines 86-88): *“Finally, larger GWAS on lipids or CRP could increase the number of pleiotropic loci in future and further expand our knowledge on the link between mechanisms of inflammation and cardiometabolism.”*

2) The term "pleiotropy" has numerous meanings. The meaning used in the article is not entirely clear. It seems to me that the pleiotropy here means the same "causal" variant associated with multiple traits (based on colocalization analysis). The mechanism of the association should be better illustrated. Is the observed multi-trait association induced by the causal relationship between traits, or due to the presence of a confounder that is simultaneously correlated with both traits, etc. Using a method such as Stephens 2012 PLOS ONE can be more informative for this purpose.

Reply: Thank you for your suggestion. We agree with the reviewer that several types of pleiotropy could be observed. We defined here pleiotropy the association between

a single genetic variant with multiple traits or phenotypes. This is referring to 'statistical pleiotropy' and includes situations of both biological (horizontal) or mediated (vertical) pleiotropy. We agree that this distinction is important to deeper understand the underlying pathophysiological mechanisms and biological pathways. However, the exact nature of pleiotropy is very challenging to be confirmed, and the interpretation should always be done with caution. We now acknowledge this limitation (p.9-10, lines 70-80) *"Fifth, we defined pleiotropy as the presence of statistically significant associations between a genetic variant and more than one phenotype. Our definition therefore refers to 'statistical pleiotropy', and includes situations of horizontal pleiotropy (for example, one SNP directly influences multiple phenotypes), and situations of vertical pleiotropy where statistical associations to multiple traits are induced via causal effects of one trait on another or via a third common factor¹⁵. Although detecting shared genetic associations between two or more traits suggest that possibly there are pleiotropic genes involved in the biological pathways of all the associated traits, still, this does not necessarily mean that the traits share biological pathways, as the pleiotropic genes could affect the traits independently via different pathways, or they could even be expressed in different tissues in response to different signalling^{21,22}."*

REVIEWERS' COMMENTS

Reviewer #1 (Remarks to the Author):

The authors have answered my questions. I have no further concerns.

Reviewer #2 (Remarks to the Author):

I consent with the author that the GLGC datasets were not available and thus do not insist on the authors redoing the analysis. I endorse the publication of the article.

REVIEWERS' COMMENTS

Reviewer #1 (Remarks to the Author):

The authors have answered my questions. I have no further concerns.

Reply: We thank the reviewer for suggesting the publication of our manuscript

Reviewer #2 (Remarks to the Author):

I consent with the author that the GLGC datasets were not available and thus do not insist on the authors redoing the analysis. I endorse the publication of the article.

Reply: We thank the reviewer for suggesting the publication of our manuscript